# Artificial Intelligence Applications in Cardiovascular Magnetic Resonance Imaging: Are We on the Path to Avoiding the Administration of Contrast Media?

**DOI:** 10.3390/diagnostics13122061

**Published:** 2023-06-14

**Authors:** Riccardo Cau, Francesco Pisu, Jasjit S. Suri, Lorenzo Mannelli, Mariano Scaglione, Salvatore Masala, Luca Saba

**Affiliations:** 1Department of Radiology, University Hospital of Cagliari, 09042 Monserrato, Italy; riccardocau00@gmail.com (R.C.); fra.pisu1@gmail.com (F.P.); 2Stroke Monitoring and Diagnostic Division, AtheroPoint™, Roseville, CA 95661, USA; jsuri@comcast.net; 3IRCCS SYNLAB SDN S.p.A., 80143 Naples, Italy; mannellilorenzo@yahoo.it; 4Department of Radiology, University Hospital of Sassari, 07100 Sassari, Italy; mscaglione@uniss.it (M.S.); samasala@uniss.it (S.M.)

**Keywords:** cardiovascular imaging, non-contrast images, AI, CMR

## Abstract

In recent years, cardiovascular imaging examinations have experienced exponential growth due to technological innovation, and this trend is consistent with the most recent chest pain guidelines. Contrast media have a crucial role in cardiovascular magnetic resonance (CMR) imaging, allowing for more precise characterization of different cardiovascular diseases. However, contrast media have contraindications and side effects that limit their clinical application in determinant patients. The application of artificial intelligence (AI)-based techniques to CMR imaging has led to the development of non-contrast models. These AI models utilize non-contrast imaging data, either independently or in combination with clinical and demographic data, as input to generate diagnostic or prognostic algorithms. In this review, we provide an overview of the main concepts pertaining to AI, review the existing literature on non-contrast AI models in CMR, and finally, discuss the strengths and limitations of these AI models and their possible future development.

## 1. Introduction

Cardiovascular magnetic resonance (CMR) offers a comprehensive evaluation of cardiovascular diseases and, according to recent guidelines, is a rapidly expanding non-invasive imaging modality [1,2]. In addition to allowing an accurate measurement of ventricular volume and function, CMR offers the capability to visualize myocardial tissue abnormalities [3]. Indeed, CMR has proven to be a crucial non-invasive imaging modality for evaluating the presence, extent, and location of myocardial fibrosis in both ischemic and non-ischemic cardiomyopathies [4,5]. Myocardial scar imaging is typically evaluated using late gadolinium enhancement (LGE) [4,5]. Gadolinium-based contrast agent freely distributes in extracellular space. However, in cases of myocardial damage, it can also enter the intracellular space through ruptured cell membranes, resulting in a delayed washout of gadolinium from the affected myocardium compared to healthy myocardium [6]. LGE imaging allows for discrimination between ischemic and non-ischemic etiologies. In addition, LGE-CMR provides important prognostic information [5,6,7,8,9].

Currently, CMR with LGE is necessary to identify myocardial scar. In clinical practice, some patients are not eligible to contrast media administration due to allergies or kidney disease. In addition, LGE sequences are time-consuming, requiring a 10–15 min time delay after contrast media administration for contrast retention in myocardial scar tissue [10,11]. Therefore, alternative non-contrast CMR tools are useful in clinical practice to improve the widespread diffusion of CMR and reduce the costs of cardiovascular healthcare.

Artificial intelligence (AI) is a broad field of computer science that is capable of performing tasks associated with human-like intelligence. AI in cardiovascular imaging has experienced a growing development [12,13,14,15,16,17,18,19]. AI-based approaches have proven useful in different CMR areas, including automated image acquisition, reconstruction, and analysis, as well as in providing diagnostic and prognostic information [12]. Recently, the field of non-contrast CMR examinations emerged through the support of AI models, with promising results. Several papers have reviewed the role of AI in CMR [13,14,15,20], or discussed the application of AI in reducing or eliminating contrast media administration in computed tomography, or in other organs beyond the heart [21,22,23,24,25]. However, to the best of our knowledge, none of the previous works have specifically focused on non-contrast AI models in CMR.

Given the exponential increase in CMR examinations in recent years, wider availability of sustainable, faster, and cheaper CMR, as well as acceleration in image acquisition, may undoubtedly provide benefits in “real life” clinical practice.

In this review, we discuss some basic concepts of AI models in CMR and provide an overview of the existing literature on non-contrast AI models in CMR, highlighting the benefits of their application in clinical practice. Lastly, we discuss the current limitations of these AI models and their possible future development.

## 2. Concepts of AI

AI is an umbrella term encompassing various techniques that allow machines to learn from experience and replicate human thought processes for solving a wide range of tasks [12,13,14,19,26]. Several approaches to AI exist, with machine learning (ML) being one of them. ML refers to the ability of AI systems to acquire knowledge about a domain by extracting patterns from raw data, without being explicitly programmed to do so. A plethora of learning algorithms exist, and each allows the creation of a model of a certain phenomenon of interest by learning through experience. In other words, the algorithms are given examples as input, and they subsequently output a learned function of these inputs. Such models can then be used to make predictions about new unseen data [27]. Typically, the features fed to ML algorithms are task-specific and need to be either handcrafted or created by other algorithms. Moreover, knowing in advance which features are required for a given task can be quite challenging. Deep learning (DL), a type of ML, solves this problem by enabling the algorithms themselves to learn and extract relevant features [14,27] (Figure 1).

This paradigm is known as representation learning and is at the foundation of DL [26]. By representing complex concepts as deep hierarchies of simpler concepts, it is possible to discover and learn high-level, complex features from simpler ones [26]. The ability of the paradigm to automatically extract features from unstructured data without the need for human intervention gradually led to the abandoning of traditional processing methods in favor of deep artificial neural networks (DNNs; also known as feedforward neural networks or multilayer perceptrons).

### 2.1. Taxonomy of ML Tasks

Machine learning tasks can be loosely categorized into different classes based on what type of experience the algorithms are allowed to have during the training process (Figure 2) [27].

#### 2.1.1. Supervised Learning

In supervised learning, the learning algorithms have access to labels (such as the global longitudinal strain of the left ventricle) for each sample in the dataset (such as cine-CMR). This labeled data, which is also referred to as ground-truth data, allows the algorithms to learn the mapping from input data to ground truths. By iteratively updating the algorithm’s learned parameters to minimize a particular cost function, and thus best match its predictions with the provided labels, the training process aims at improving the accuracy and generalizability of the algorithm. The accuracy of the learned mapping is then evaluated on test data, which is unseen during the training stage. In medical image analysis, the input to these algorithms is often a set of 2D images or 3D volumes, or even sequences of 2D or 3D images in time, such as a series of images of the entire cardiac cycle in cine-CMR. The learned output depends on the specific medical task (Figure 3).

##### Taxonomy of Supervised Tasks

In regression tasks, the algorithm produces a continuous output (such as the ejection fraction values of the left ventricle); in classification tasks, the output is a set of discrete labels (such as the different levels of cardiac contractility). Other tasks include localization tasks, in which the algorithm estimates the coordinates in the image space of bounding boxes around regions of interest (such as anatomical or pathological structures); segmentation tasks, in which the algorithm outputs a class label for each pixel or voxel in the image or volume, typically distinguishing between the background and one or more foreground regions of interest, such as the left ventricle and atrium in non-contrast cine-CMR; acquisition tasks, in which DNNs can be leveraged to adjust imaging parameters in real-time based on patient-specific characteristics, such as heart rate and anatomy, in order to improve spatial resolution, reduce motion artifacts, and enhance the overall image quality; and finally, reconstruction tasks, in which DNNs can be trained to reconstruct high-resolution cardiac images from undersampled k-space data, acquired with accelerated imaging techniques such as compressed sensing [28].

In recent AI research, a notable emerging trend revolves around the seamless integration of imaging data with diverse information obtained from different sensors, all pertaining to a common phenomenon. This paradigm, known as multimodal learning, aims to leverage these heterogeneous data in a complementary manner to facilitate the learning of complex tasks [29]. These data can be clinical information (such as patient demographics and medical history), genetic or molecular data (such as molecular profiles of patients), behavioral or lifestyle data (such as habits, diet, physical activity, or environmental exposures), longitudinal data regarding clinical outcomes, or disease progression and histopathological data (such as information obtained from tissue samples).

Creating a dataset for training a supervised ML algorithm involves the production of ground-truth output data. This crucial step involves the meticulous annotation of regions of interest or the assessment of quantitative measures by radiologists or medical experts. Manual tracing of masks around complex structures in 3D volumes, such as cardiac structures for segmentations tasks or evaluating atrial and ventricle functions or strain values, is particularly demanding. It requires the annotator to meticulously delineate borders across multiple slices, underscoring the labor-intensive nature of the process. Consequently, generating high-quality labeled datasets in the medical domain requires both expertise and a significant investment of time. Moreover, DL approaches trained in a supervised manner often necessitate a substantial amount of annotated data, which can pose a limitation to their applicability to specific tasks.

#### 2.1.2. Unsupervised Learning

In unsupervised learning, the algorithms are exposed to large volumes of unlabeled examples, with the goal of learning useful properties about the structure of the data. Common tasks include estimating the probability distribution underlying the data generation process, clustering the data into related subgroups, or learning to denoise noisy data.

#### 2.1.3. Hybrid Paradigms

Hybrid paradigms that lie between supervised and unsupervised learning include semi-supervised, weakly-supervised, and self-supervised learning. In weakly-supervised learning, the (possibly inaccurate) supervision comes in the form of noisy labels that may not always be ground truths, such as partial masks around regions of interest (Figure 2); this can be due to imprecise or weary labeling, or there may be inherent complexities in categorizing some examples. Semi-supervised learning, also known as incomplete supervision, refers to a learning paradigm where only a small portion of the dataset is annotated; the goal is to leverage domain knowledge and subject matter expertise to exploit the unlabeled data and use it in conjunction with labeled data to learn good feature representations. Finally, in self-supervised learning, only unlabeled data is available. These algorithms frame the learning objective in such a way that supervision comes from the data itself, for example, by deriving pseudolabels from intrinsic attributes of the data. Through the definition and subsequent resolution of a pretext or self-supervised task, it is possible to identify intermediate feature representations that can be used to solve other downstream tasks, such as image classification or object detection, with very few annotated examples [15,20].

### 2.2. Convolutional Neural Networks and the U-Net Architecture

In recent years, DL has experienced tremendous growth thanks to advances in technology and hardware, big data availability, and more powerful model architectures. DL algorithms have achieved state-of-the-art results in fields such as computer vision, surpassing other established methods on several image analysis benchmarks and across multiple domains, including medical imaging. Promising DL-based approaches have been proposed for a variety of image processing and analysis tasks, such as reconstruction methods for reducing scan time, noise, and artifact reduction, resolution improvement, and localization and segmentation of regions of interest, such as the left atrium in cine-CMR [12,30].

Most DL approaches for medical image analysis are based on convolutional neural networks (CNNs). CNNs are a DL algorithm designed to analyze visual data such as images and videos. They consist of multiple layers of interconnected artificial neurons known as convolutional layers. These layers employ filters to process input data, effectively capturing spatial relationships and patterns. The filters execute convolutions by sliding across the input data, multiplying values, and generating feature maps that emphasize significant features. For detection, localization, and segmentation tasks, one of the most popular CNN-based architectures is the U-Net [31]. The U-Net architecture (and its variants) consists of a U-shaped structure comprising two interconnected paths. The contracting path enables the extraction of increasingly abstract, fine-grained features while preserving both local and global contextual information. This is achieved through a series of downsampling operations that progressively reduce the spatial resolution of the image. On the other hand, the expanding path involves upsampling the image back to its original spatial resolution by utilizing skip connections. These connections establish direct links between corresponding blocks at the same hierarchical level, facilitating the integration of high-resolution features. This U-shaped architecture allows for effective information flow and combines local details with a global context, making it suitable for various image analysis tasks. For instance, a U-Net could be trained to identify and segment the left atrium for the automated quantification of left atrium volume in routine long-axis cardiac cine images. Likewise, U-Nets can be used for image quality enhancement, where the network learns to denoise corrupted images. For such tasks, a training set comprised of corrupted–uncorrupted image pairs is required to train the model. After pixel-wise alignment, the model is learned by minimizing the error function between the input and network-generated images. While this approach would be ideal, acquiring such real pixel-aligned image pairs in routine clinical practice is difficult for various reasons, including pixel misalignment due to motion artifacts [12]. Semi and self-supervised learning approaches have the potential to overcome these limitations thanks to their ability to make the most out of unlabeled data.

### 2.3. AI-Based Diagnosis and Prognosis Prediction

Clinical prediction models have become increasingly important in modern clinical practice, providing valuable information to healthcare professionals and patients regarding outcome risks. These models facilitate decision-making processes and ultimately improve health outcomes [32]. Diagnostic prediction models estimate the probability of current health conditions being present at the individual level, while prognostic models estimate the likelihood of developing a medical outcome within a specified time frame, such as the probability of major adverse cardiac events over a five-year period [33,34].

In recent years, AI-based prognostic models have shown promise in providing improved stratification of prognosis compared to traditional clinical prognostic scores. These approaches require time-to-event data and may involve classical ML algorithms trained on manually derived features from experienced radiologists, as well as demographic, clinical, and traditional cardiovascular risk factors [35]. Alternatively, DL algorithms can operate directly on images or volumes, automatically extracting relevant features to make a prognostic prediction. DL is commonly employed to isolate regions of interest, such as the left ventricle in short-axis cine-CMR images. These segmentations can be utilized to compute quantitative measures, like global circumferential strain, which can then be incorporated into traditional risk-score models (e.g., the Cox proportional hazards model). Alternatively, DL can directly use the segmentations to make individualized prognosis predictions.

AI-based methods, especially DL methods, have demonstrated remarkable scalability to handle large datasets and computationally-intensive tasks, making them highly suitable for various tasks in the field of medical image analysis. Specifically, in the CMR domain, DL approaches have showcased their capabilities in addressing a diverse range of tasks, making it a valuable tool for advancing CMR research and improving diagnostic accuracy and patient care.

## 3. AI Applications in Non-Contrast Cardiovascular Magnetic Resonance

Contrast media administration is fundamental in CMR imaging, ranging from diagnosis to prognosis [36,37,38,39,40,41,42]. Indeed, contrast enhancement is essential for evaluating the presence of myocardial fibrosis using late gadolinium enhancement sequences [38,43,44]. The size, location, and extent of myocardial fibrosis allow discrimination between ischemic and non-ischemic etiologies and are independent risk factors for adverse outcomes [5,8,45,46,47,48,49,50,51].

Despite the crucial diagnostic and prognostic role of the LGE sequence, some patients are not eligible to contrast media administration due to allergic reactions and nephrotoxicity [11,52]. In addition, the application of a rapid CMR acquisition protocol without contrast media can reduce examination time, allowing more widespread dissemination of CMR scans with less cost.

Recently, parametric mapping techniques have been introduced in the daily clinical practice routine, allowing for the discrimination of pathologic myocardium from healthy tissue [7,36,53,54,55]. Various studies investigated the potential role of parametric mapping to assess myocardial scar without the use of gadolinium contrast in comparison with LGE images [56,57]. However, T1 and T2 mapping require additional sequences that extend the CMR acquisition time, and the relaxation time is field-strength and scanner specific, leading to inter-center variability and limiting a widespread clinical utility [58].

In the last years, AI is rising as a leading component in cardiovascular medicine [12,13,19] and some studies have attempted to generate AI models without contrast media administration. Table 1 summarizes previous studies regarding the application of non-contrast artificial intelligence models in CMR.

### 3.1. Ischemic Cardiomyopathy

CMR has evolved as an essential tool for assessing the prognosis after myocardial infarction as endorsed by the Society for Cardiovascular Magnetic Resonance 2020 position paper [76]. Indeed, the presence of LGE allows for discrimination between viable and non-viable myocardium and the likelihood of recovery after revascularization [44,77,78,79]. Moreover, the presence of LGE is a predictor of major adverse cardiovascular events, independent of left ventricle ejection fraction [47].

Several researchers have explored the potential role of non-contrast cine-CMR images as an alternative to late gadolinium enhancement (LGE)-CMR images to assess myocardial infarction location and size without gadolinium injection [59,60,61,62,63,64,66]. Baessler et al. investigated the application of radiomics features using non-contrast cine-CMR to differentiate ischemic scar and normal myocardium. Using multiple logistic regression models, the authors demonstrated an area under the receiver-operating characteristics curve (AUC) of 0.93 and 0.092 in diagnosing large and small myocardial infarction on cine-CMR, respectively [59]. In the retrospective study of 72 patients (52 with myocardial infarction and 20 healthy control patients), an ML-based model was developed using radiomics features for the distinction of myocardial infarction tissue and viable myocardium on non-contrast cine-CMR images. The authors reported optimal performance for the logistic regression model with an AUC of 0.93 ± 0.03, an accuracy of 0.86 ± 0.05, a recall of 0.87 ± 0.1, a precision of 0.93 ± 0.03, and an F1 Score = 0.90 ± 0.04, and for the support vector machine model with an AUC of 0.92 ± 0.05, an accuracy of 0.85 ± 0.04, a recall of 0.92 ± 0.01, a precision of 0.88 ± 0.04, and an F1 Score of 0.90 ± 0.02, respectively [60].

A pixel-wise tissue identification approach using a DL architecture was recently proposed from non-contrast CMR images [67]. The AI-based model consisted of three connected functional layers, namely heart localization, which automatically delineates the LV, the motion feature extraction to build motion features through intensity changes between adjacent images to evaluate the motion of each pixel, and the fully-connected discriminative layers to predict tissue identification in each pixel. The authors validated the proposed pipeline in 165 cine-CMR imaging datasets, reporting high accuracy (pixel-level: 95.03%) and consistency (Kappa statistic: 0.91; Dice: 89.87%) in comparison to the ground truths manually segmented LGE images [67].

Xu et al. proposed a contrast-free deep spatiotemporal generative adversarial network to simultaneously segment and quantify myocardial infarction from cine-CMR images [68]. The proposed AI-based model used a conditional generative adversarial network DL approach, achieving a pixel classification accuracy of 96.98%, and the mean absolute error of the infarction size was 17.15 mm^2^ [68].

Larroza et al. trained a support vector machine classifier to investigate the ability of texture analysis using cine-CMR images to discriminate between infarcted nonviable, viable, and remote segments. The authors demonstrated that non-viable segments can be detected on non-contrast cine-CMR images using texture analysis, with an AUC of 0.849 and a sensitivity of 92% [62].

Chen et al. developed an ML model that combined physiological, clinical, and paraclinical features to evaluate the severity of myocardial infarction in 150 patients. The proposed model revealed with high accuracy the presence of infarction, persistent microvascular dysfunction, and the percentage of infarcted myocardium, demonstrating a mean error of 0.056 and 0.012 for the quantification, and 88.67 and 77.33% for the classification accuracy of the state of the myocardium [65]. In a retrospective study of 272 patients with diagnoses of myocardial infarction (*n* = 108) and healthy controls (*n* = 164), an AI-based model was investigated to predict post-contrast information (i.e., presence, location, and/or extent of MI scar) from non-contrast data [69]. The authors described a pipeline and explored two different approaches, namely, segmentations and classification models. For the first approach, a U-Net DL model was investigated to discover the extent and location of the myocardial scar from non-contrast cine-CMR images [69]. For the classification model, a ResNet50 was used to discriminate between ischemic from non-ischemic patients given the non-contrast cine-CMR images [69]. Finally, two supervised ML methods, namely, the support vector machines (SVM) and the decision tree (DT) methods were developed for classifying myocardial infarction patients and healthy subjects from non-contrast cine-CMR images. The SVM model achieved accuracy, F1, and precision scores of 0.68, 0.69, and 0.64, respectively. Conversely, the DT models achieved accuracy, F1, and precision scores of 0.62, 0.63, and 0.72, respectively [69]. The main limitations of the previously discussed studies were the small cohorts of patients enrolled from a single center, the retrospective data analysis, and the lack of external validation.

Zhang et al. combined cine-CMR images and native T1 mapping to produce LGE-like images using a novel AI approach, namely virtual native enhancement. This approach used a DL model to enhance the imaging signal in native T1 mapping and cine images. The virtual native enhancement was compared with LGE images using linear regression, Pearson correlation, and intraclass correlation coefficients. In addition, a histological comparison was performed in the porcine model of myocardial infarction. The authors developed a non-contrast DL model from 843 patients with previous myocardial infarction using 775 patients for development and 68 patients for testing. Virtual native enhancement demonstrated a strong correlation with LGE in quantifying scar size (R, 0.89; intraclass correlation coefficient, 0.94) and transmurality (R, 0.84; intraclass correlation coefficient, 0.90), achieving an accuracy of 84% for detecting scars with a specificity of 100% and a sensitivity of 77%, and excellent visuospatial agreement with the histopathological porcine model [64]. In their prospective study, Zhang et al. enrolled a large cohort of patients with myocardial infarction from a single center. However, they did not evaluate the DL model using independent testing, which represents a limitation of their study [64].

### 3.2. Non-Ischemic Cardiomyopathy

Similar AI-based models have been proposed in non-ischemic cardiomyopathies [70,71,72,73]. A non-contrast T1 mapping CMR ML-based model using radiomics for the detection of myocardial tissue alterations in hypertrophic cardiomyopathy was developed by Baeßler et al. In their retrospective study, the authors analyzed radiomics features in 32 patients with known hypertrophic cardiomyopathy in comparison with 30 healthy patients. The proposed ML model achieved an AUC of 0.95, with a diagnostic sensitivity of 91% and a specificity of 93% [71]. The application of an AI-based model combining imaging, clinical, and demographic data has been shown to be an effective tool in the differential diagnosis of various cardiovascular diseases.

Cau et al. developed an ML-based model integrating non-contrast CMR parameters and demographic factors to identify patients with Takotsubo cardiomyopathy in subjects with acute chest pain. The authors retrospectively enrolled three groups of patients (patients with Takotsubo cardiomyopathy, patients with acute myocarditis, and healthy subjects) and tested five different tree-based ensemble learning algorithms, namely Random Forest, Extremely Randomized Trees, Bagging of Trees, Adaptive Boosting, and Extreme Gradient Boosting. The Extremely Randomized Trees ML algorithm showed a sensitivity of 92% (95% CI 78–100), specificity of 86% (95% CI 80–92), and AUC of 0.94 (95% CI 0.90–0.99) in diagnosing Takotsubo cardiomyopathy. In addition, the proposed model outperformed clinical reader diagnoses with an average increase in AUC of 0.42 (80%), a sensitivity of 0.08 (10%), and a specificity of 0.618 (257%) in a shorter analysis time (0.26 s vs. 560 s) [73]. Similarly, Eckstein et al. developed a supervised ML model combining right atrial, left atrial, and right ventricular strain parameters and cardiac function to identify patients with cardiac amyloidosis using multiple machine learning classifier algorithms such as the k-nearest neighbor, both linear and non-linear support vector machines, and decision trees. Non-linear support vector machine using a kernel radial basis function demonstrated the highest accuracy rate among all ML algorithms (90.9%), with an AUC of 0.996 [74]. The main limitations of the previously discussed studies were the retrospective data analysis, the small cohorts of patients enrolled from a single center for the training cohorts, and the lack of an evaluation using independent testing. A CMR virtual native enhancement has also been proposed for non-ischemic cardiomyopathy using a DL-based model by Zhang et al. The AI model used multiple streams of convolutional neural networks to enhance contrast and signals within the native T1 maps and cine-CMR images obtaining an LGE-equivalent image. The virtual native enhancement was developed from the multicenter Hypertrophic Cardiomyopathy Registry using 1075 patients for the development and 121 patients for the test. The authors reported high agreement between virtual native enhancement and LGE in the visuospatial distribution and quantification of lesions [70]. Cine-CMR images analyzed with DL also showed their potential prognostic role in the recent CERTAINTY study (CinE caRdiac magneTic resonAnce to predIct veNTricular arrhYthmia). In this study, the authors evaluated a DL model for ventricular arrhythmia risk prediction from non-contrast cine-CMR of 350 heart failure patients with both ischemic and non-ischemic etiologies and created a cine risk score. The proposed score remained significantly associated with ventricular arrhythmia after adjustment in the multivariable regression analysis (Model I: adjusting for sex, type of cardiomyopathy, use of diuretics, and hsCRP; HR 3.24, *p*  =  0.005. Model II: further adjusting for LVEDI, LV ejection fraction, LV LGE gray zone, LA maximum volume index, and LA total emptying fraction; HR 2.67, *p*  =  0.027) [75]. Although it delivered promising prognostic results, the CERTAINTY study has intrinsic limitations, including the lack of an external validation cohort and the small sample size obtained from a single institution for the training cohort.

## 4. Current Limitations and Future Developments

The translation of non-contrast AI models into real-world clinical practices faces significant challenges. The “black box” nature of AI algorithms, particularly DL models, hampers interpretability and explainability. The reproducibility of results across different settings is also a crucial concern. Data heterogeneity, encompassing variations in imaging protocols and patient demographics, poses a challenge to generalizability. Ethical and legal issues, including privacy and data security, must be carefully addressed. Overcoming these challenges is essential for the successful integration of non-contrast AI-based solutions in clinical practice.

### 4.1. Lack of Algorithms Transparency and Quality Control

The lack of transparency and interpretability often creates ambiguity regarding what data is used and the interactions of the underlying variables that lead to specific outputs, thereby limiting the applicability of AI models in clinical practice without adequate insight or explanation. Furthermore, as the size of DL models increases, it becomes progressively challenging to explain the individual contributions of the network’s elements to the generated predictions.

### 4.2. Data Heterogeneity and Concerns with Validation and Testing

Developing robust and generalizable AI models for CMR image analysis also requires a substantial amount of annotated data from diverse centers, along with highly specialized software and infrastructure. It is essential to have a large and homogeneous dataset that encompasses images from various centers to ensure the model’s reliability and generalizability. In addition, when such a dataset is available, it is important to examine and separate it into training, validation, and testing sets. This division allows for an impartial evaluation of the final models, ensuring that the assessment is not biased [80]. Furthermore, it is of fundamental importance to repeat the division with different randomizations or employ consolidated techniques for internally evaluating the generalization capability of the models, such as repeated stratified cross-validation or leave-one-site-out testing [81,82].

### 4.3. Ethical and Legal Issues

The use of AI in healthcare raises important ethical and legal considerations. The main ethical concerns about the application of AI in healthcare include data privacy and security, transparency and safety, and algorithm fairness and bias. The development of large and homogeneous datasets primarily raises concerns about data privacy and security. Notable apprehensions arise regarding the collection and utilization of patient data, including the possibility of its employment in undisclosed manners by entities distinct from the individual from whom the data was obtained. Additionally, there exists the potential for the information gathered for and by AI systems to be vulnerable to hacking attempts [83,84]. Regulatory bodies such as the Food and Drugs Administration and the European Union Medical Device Regulation have initiated efforts to ensure transparency and safety in data handling, privacy, processing, and data sharing [85,86,87]. Another issue is the “fairness” of the AI model. Indeed, an imbalance in the training data can lead to notable variations in the performance of AI-based models across various sex and racial groups, potentially exacerbating disparities in healthcare.

### 4.4. Future Development

Controlled trials are needed to assess the application of non-contrast AI models across different centers and patient groups to avoid unbalanced sex representation and low representation of minority communities in order to minimize bias. Multicenter and multivendor studies are necessary to evaluate the effectiveness of non-contrast AI-based models in real-world clinical practice. It is crucial to compare the performance of the AI algorithm with expert evaluations of “gold standard” CMR images, such as LGE images, by conducting multireader, multicase studies involving multiple experienced radiologists [88]. This approach allows for a comprehensive analysis of the model’s performance, benchmarking it against the collective expertise of human specialists. Furthermore, conducting external validation using data from different centers and acquisition scanners enhances the reliability and generalizability of the AI model’s findings. Similarly, ensuring a robust quality control process is vital for the clinical translation of AI-based approaches, leveraging the availability of open-source models and datasets. Non-contrast AI models may represent a potential evolution in CMR imaging, allowing faster examinations at significantly reduced lower costs. By implementing effective quality control measures, such as rigorous statistical validation of models, the clinical integration of AI-based approaches can be facilitated, leading to improved efficiency, cost-effectiveness of CMR imaging practices, and overall patient care.

In addition, it would be necessary to develop a privacy protection algorithm that integrates encryption and AI techniques for achieving secure and generalizable non-contrast AI models. A potential strategy for safeguarding patient privacy is to keep the datasets within the firewall of each organization and transfer only the algorithms or trained parameters to each site.

## 5. Conclusions

The adoption of non-contrast AI-based models holds great potential to enhance daily clinical practice in cardiovascular healthcare by reducing costs and facilitating wider access to CMR examinations. In particular, these AI models may be particularly useful in patients with kidney diseases or those not eligible for contrast media administration due to allergies. Application of these AI technologies may have the potentiality to significantly assist clinicians in “real life” clinical practice routine, facilitating the clinical workflow, diagnosis, and prognosis in ischemic and non-ischemic cardiomyopathy.

However, the applicability of non-contrast CMR approaches is currently limited due to the challenges related to heterogenous datasets from different centers, the lack of transparency and explainability of many AI algorithms, and insufficient external validation in existing studies. To expedite the clinical application of these models, it is crucial to design controlled trials carefully, addressing these limitations and ensuring robustness and reliability in their evaluation. Future prospective, multicenter, multireader, and paired validation studies are needed to accelerate the translation of non-contrast AI-based models into effective tools for clinical decision-making in CMR imaging.

## Figures and Tables

**Figure 1 diagnostics-13-02061-f001:**
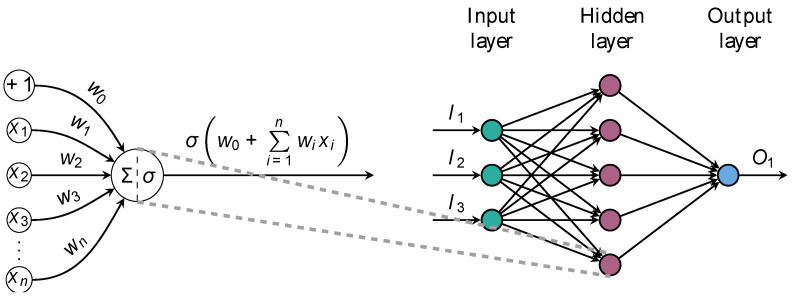
Anatomy of a feedforward artificial neural network. Illustration of a one-layer fully-connected network (FCN) architecture. The network consists of an input layer with three units, exemplified here by age, smoking status, and left ventricle ejection fraction. Hidden neurons receive inputs, compute weighted sums, and pass through nonlinear activation functions to produce outputs. The network maps process data to output units, providing probabilities of class membership or estimating numeric quantities of interest. The primary objective of this architecture is to learn complex nonlinear mappings between input and ground-truth data.

**Figure 2 diagnostics-13-02061-f002:**
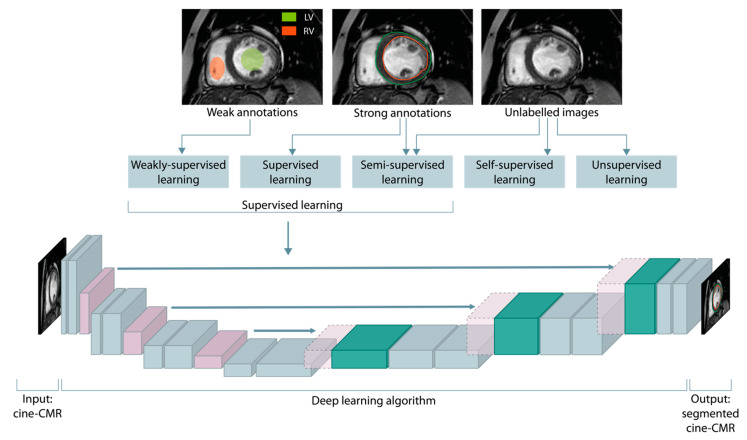
Taxonomy of machine learning algorithms based on the presence and type of supervision. The supervision is commonly provided through weak or strong ground truths, encompassing regions of interest annotations, estimation of numeric quantities, and assignment of discrete class labels. Below, a U-shaped deep neural network for a supervised segmentation of the endocardium (red contour) and epicardium (green contour) of the left ventricle is shown.

**Figure 3 diagnostics-13-02061-f003:**
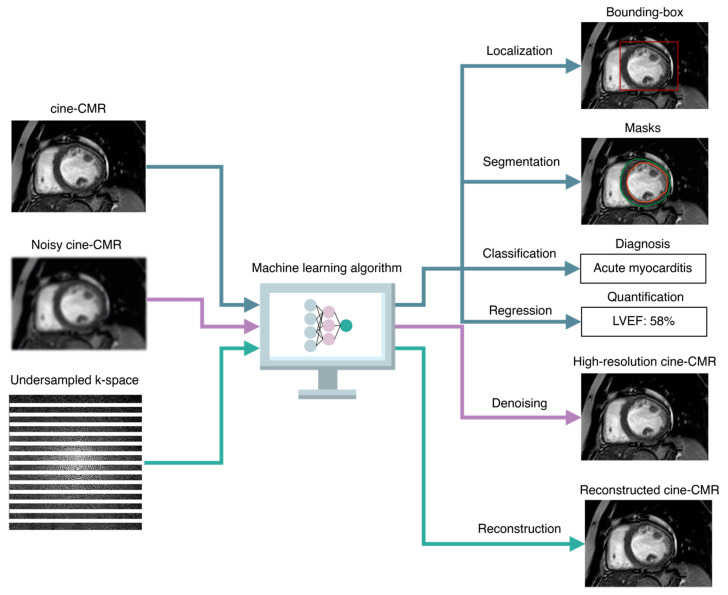
Taxonomy of AI-based image analysis tasks. Common supervised image analysis tasks include classification and regression tasks, in which the AI algorithms are trained to predict discrete class membership likelihoods and numeric quantities, respectively. Other tasks that are exclusively related to images include localization, in which the algorithms predict the coordinates of bounding boxes around regions of interest (such as the left ventricle in cine-CMR), segmentation, in which each pixel or voxel in the image is assigned to either the background or one or more classes of interest (such as the epicardium and endocardium contours), denoising, which consists of producing high-resolution images or volumes (such as cine-CMR images over the cardiac cycle) from noisy inputs, and reconstruction, in which the algorithms are taught to reconstruct high-quality cardiac images or volumes from undersampled k-spaces. LVEF indicates left ventricle ejection fraction.

**Table 1 diagnostics-13-02061-t001:** Previous CMR studies about non-contrast AI models in ischemic and non-ischemic cardiomyopathy.

Authors	Years	Number of Patients	Variables	Results
Baessler et al. [59]	2018	120	Radiomics features of cine-CMR images	The model demonstrated an AUC of 0.93 and 0.092 to diagnose large and small myocardial infarction on cine-CMR.
Avard et al. [60]	2022	72	Radiomics features of cine-CMR images	The authors reported optimal performance for the logistic regression model with an AUC of 0.93 ± 0.03, an accuracy of 0.86 ± 0.05, a recall of 0.87 ± 0.1, a precision of 0.93 ± 0.03, an F1 Score = 0.90 ± 0.04, and for the support vector machine model with an AUC of 0.92 ± 0.05, an accuracy of 0.85 ± 0.04, a recall of 0.92 ± 0.01, a precision of 0.88 ± 0.04, and an F1 Score of 0.90 ± 0.02, respectively.
Larroza et al. [61]	2017	44	Radiomics features of cine-CMR images	Radiomics analysis of cine-CMR images achieved an AUC of 0.82 ± 0.06 with a sensitivity of 0.79 ± 0.10, and specificity of 0.80 ± 0.10 for differentiation of acute myocardial infarction from chronic myocardial infarction.
Larroza et al. [62]	2018	50	Radiomics features of cine-CMR images	The model demonstrated an AUC of 0.849, and a sensitivity of 92% to detect nonviable segments, 72% to detect viable segments, and 85% to detect remote segments.
Zhang et al. [63]	2019	212	Cine-CMR	The DL model showed a sensitivity of 89.8% and a specificity of 99.1%, with an AUC of 0.94.
Zhang et al. [64]	2022	843	Cine-CMR images, T1 mapping images	Virtual native enhancement demonstrated a strong correlation with LGE in quantifying scar size (R, 0.89; intraclass correlation coefficient, 0.94) and transmurality (R, 0.84; intraclass correlation coefficient, 0.90), achieving an accuracy of 84% for detecting scars with a specificity of 100% and sensitivity of 77%, and excellent visuospatial agreement with the histopathological porcine model.
Chen et al. [65]	2022	150	Physiological, clinical, and paraclinical features	The proposed model demonstrated a mean error of 0.056 and 0.012 for the quantification, and 88.67 and 77.33% for the classification accuracy of the state of the myocardium.
Goldfarb et al. [66]	2019	90	CMR water–fat separation and parametric mapping	The DL model could visualize myocardial fat deposition in chronic myocardial infarction and intramyocardial hemorrhage in acute myocardial infarction.
Xu et al. [67]	2020	165	Cine-CMR images	The proposed AI-based model achieved a pixel classification accuracy of 96.98%, and the mean absolute error of the infarction size was 17.15 mm^2^.
Xu et al. [68]	2018	165	Cine-CMR images	The proposed framework for the pixel-wise delineation of the myocardial infarction area achieved an accuracy of 95.03% and optimal consistency (Kappa statistic: 0.91; Dice: 89.87%) in comparison to the ground truths manually segmented LGE images.
Abdulkareem et al. [69]	2022	272	Cine-CMR images	The SVM model achieved accuracy, F1, and precision scores of 0.68, 0.69, and 0.64, respectively. Conversely, the DT models achieved accuracy, F1, and precision scores of 0.62, 0.63, and 0.72, respectively.
Zhang et al. [70]	2021	1196	Cine-CMR images, T1 mapping images	The authors reported high agreement between virtual native enhancement and LGE in the visuospatial distribution and quantification of lesions.
Baeßler et al. [71]	2018	32	Radiomics features of T1 mapping images	The proposed ML model achieved an AUC of 0.95 with a diagnostic sensitivity of 91% and a specificity of 93%.
Fahmy et al. [72]	2022	759	Radiomics features of cine-CMR	The DL model using radiomics data of cine-CMR images correctly identified 43% and 28% of HCM patients without scars in the internal and external datasets.
Cau et al. [73]	2022	43	CMR parameters, demographics data	The model showed a sensitivity of 92% (95% CI 78–100), specificity of 86% (95% CI 80–92), and AUC of 0.94 (95% CI 0.90–0.99) in diagnosing Takotsubo cardiomyopathy.
Eckstein et al. [74]	2022	96	CMR strain and function parameters	The supervised ML model demonstrated an accuracy of 90.9% (0.996; precision = 94%; sensitivity = 100%; F1 Score = 97%) to identify cardiac amiloidosis patients.
Krebs et al. [75]	2021	350	Cine-CMR images	The proposed score remained significantly associated with ventricular arrhythmia after adjustment in multivariable regression analysis.

## Data Availability

Not applicable.

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
