# Peer review of "Artificial Intelligence Applications in Cardiovascular Magnetic Resonance Imaging: Are We on the Path to Avoiding the Administration of Contrast Media?"

_diagnostics, 2023, doi:10.3390/diagnostics13122061_

Round 1

Reviewer 1 Report

This review paper provides an overview of the use of Artificial Intelligence (AI) in Cardiovascular Magnetic Resonance (CMR) imaging. The authors discuss the potential for AI to reduce the need for contrast media in imaging procedures and review existing literature on non-contrast AI models in CMR.

The article is well-organized and provides a comprehensive literature review on non-contrast AI models in CMR. The authors also provide a clear overview of key concepts related to AI and CMR.

One potential weakness is that the authors briefly mention limitations of current non-contrast AI models, they do not provide detailed recommendations for addressing these limitations. To improve their work, the authors could consider providing more detailed recommendations for addressing limitations of current non-contrast AI models.

Overall, this is a well-written and informative review paper that provides valuable insights into the use of AI in CMR imaging. The authors could consider expanding their discussion on limitations and providing more detailed recommendations for future research directions.

Author Response

This review paper provides an overview of the use of Artificial Intelligence (AI) in Cardiovascular Magnetic Resonance (CMR) imaging. The authors discuss the potential for AI to reduce the need for contrast media in imaging procedures and review existing literature on non-contrast AI models in CMR. The article is well-organized and provides a comprehensive literature review on non-contrast AI models in CMR. The authors also provide a clear overview of key concepts related to AI and CMR. One potential weakness is that the authors briefly mention limitations of current non-contrast AI models, they do not provide detailed recommendations for addressing these limitations. To improve their work, the authors could consider providing more detailed recommendations for addressing limitations of current non-contrast AI models. Overall, this is a well-written and informative review paper that provides valuable insights into the use of AI in CMR imaging. The authors could consider expanding their discussion on limitations and providing more detailed recommendations for future research directions. The authors must address all of the below concerns carefully.

Dear reviewer, we appreciate your valuable comment. We have expanded the "current limitations and future development" section and further subdivided it into relevant sub-headings. This enhancement allows for a more comprehensive exploration of the topic. Furthermore, we provided additional details and recommendations for future research directions in the revised version of the manuscript.

Reviewer 2 Report

The authors must address all of the below concerns carefully.

-          The Introduction section is not exhaustive and does not contain enough information to cover the review. It should have a comprehensive scientific background. Also at the end of the introduction, the authors should add scientific contributions to this review accurately.

-          Section 2 should be divided into sub-headings to make it more clearly to the reader.

-          It is recommended to add a comparison among essential tools for assessing the prognosis.

-          We have not seen any clear criticism of the research included in the 3.1 and 3.2 subsections.

-          The section "Current limitations and future development" requires major improvement. The selections are blurred. This section is not organized. Why was the phrase "In conclusion" added, page 12, line 381? This section requires writing in a clear and sequential manner.

-          Also, the conclusion section does not contain clear conclusions. This section should be rewritten.

-          There is an exaggerated self-citation.

-          This research in its current form is not sufficient. Since this research is a review research, it is the duty of researchers to include the difference of this review from previous reviews. Also, some comparisons and analyzes of previous studies should be added on the topic of the review.

-          Table: Table 1 does not include a title. Also, this table requires improved column alignment.

-          English Writing: This paper needs minor proofreading. There are some of grammatical, spelling and typos mistakes. The authors have to thoroughly scrutinize the paper.

-          List of References: The number of references is sufficient. However, some references do not contain enough information such as [15] … etc. Some references are not well formatted such as [40], [43] … etc. Some search names in the reference list begin an uppercase letter for each word (such as [3], [4] ... etc.) and others use only an uppercase letter in the first word (such as [3], [6] … etc.), authors should standardize style. This research requires extensive improvement to remove all problems related to the reference list. 

 English Writing: This paper needs minor proofreading. There are some of grammatical, spelling and typos mistakes. The authors have to thoroughly scrutinize the paper.

Author Response

The Introduction section is not exhaustive and does not contain enough information to cover the review. It should have a comprehensive scientific background. Also at the end of the introduction, the authors should add scientific contributions to this review accurately.

Dear reviewer, thank you for your comment. We expanded the “introduction” section providing more scientific background about the topic discussed.

-          Section 2 should be divided into sub-headings to make it more clearly to the reader.

We agree with the reviewer. In the revised version of the manuscript, the section 2 has been divided into sub-headings concerning taxonomy of machine learning tasks (supervised, unsupervised and hybrid approaches), convolutional neural networks and the commonly employed U-Net architecture.

Here’s the new structure of Section 2, “Concepts of AI”:

  • Taxonomy of ML tasks
    • Supervised learning
      • Taxonomy of supervised tasks
    • Unsupervised learning
    • Hybrid paradigms
  • Convolutional neural networks and the U-Net architecture
  • AI-based diagnosis and prognosis prediction (New, see next reviewer’s comment)

-          It is recommended to add a comparison among essential tools for assessing the prognosis.

We thank the reviewer for the recommendation. We added a sub-section in Section 2 called “AI-based diagnosis and prognosis prediction” where we address the role of Artificial Intelligence-based methods on predicting subject-level prognosis. Lines 238-266 of the revised manuscript.

-          We have not seen any clear criticism of the research included in the 3.1 and 3.2 subsections.

Dear reviewer, thank you for your comment. We added and discussed the limitations of the reported CMR studies in both ischemic (section 3.1) and non-ischemic cardiomyopathy (section 3.2).

-          The section "Current limitations and future development" requires major improvement. The selections are blurred. This section is not organized. Why was the phrase "In conclusion" added, page 12, line 381? This section requires writing in a clear and sequential manner.

Dear reviewer, thank you for your comment. We have expanded the "current limitations and future development" section and further subdivided it into relevant sub-headings to make it clearer for the reviewer.

-          Also, the conclusion section does not contain clear conclusions. This section should be rewritten.

Thank you for pointing this out. We expanded and improved the “discussion” section in the revised version of the manuscript.

-          There is an exaggerated self-citation.

Dear reviewer, thank you for your comment. In the revised version of the manuscript, we have included only the most relevant references for the research topic.

-          This research in its current form is not sufficient. Since this research is a review research, it is the duty of researchers to include the difference of this review from previous reviews. Also, some comparisons and analyzes of previous studies should be added on the topic of the review.

Dear reviewer, thank you for your comment. We discussed the differences of our review in comparison with the previous studies in the “introduction” section. In addition, we added the limitations of the studies discussed in section 3.1 and 3.2 in the revised version of the manuscript.

-          Table: Table 1 does not include a title. Also, this table requires improved column alignment.

Thank you for pointing this out. We incorporated a title for Table I and reconfigured the column alignment in the revised version of the manuscript.

-          English Writing: This paper needs minor proofreading. There are some of grammatical, spelling and typos mistakes. The authors have to thoroughly scrutinize the paper.

Dear reviewer, we appreciate your valuable comment. We have carefully reviewed and revised the manuscript, addressing any concerns regarding the English language and making the necessary corrections accordingly.

-          List of References: The number of references is sufficient. However, some references do not contain enough information such as [15] … etc. Some references are not well formatted such as [40], [43] … etc. Some search names in the reference list begin an uppercase letter for each word (such as [3], [4] ... etc.) and others use only an uppercase letter in the first word (such as [3], [6] … etc.), authors should standardize style. This research requires extensive improvement to remove all problems related to the reference list. 

Thank you for pointing this out. We have revised the manuscript and reformatted the reference list, ensuring that each reference adheres to a standardized style

Reviewer 3 Report

Very interesting and timely article. I think it deserves publication and I am recommending accept with corrections. There are some issues that require your attention. I list these corrections below as feedback / comments, and I am looking forward to reading the updated version of this article. 

- The level of originality of the paper is high. The literature review and proposed design are suitable. The results are discussed: authors found the values and risks of new AI use cases in Cardiovascular Magnetic 2

Resonance imaging.

- I have finished reading the article and I didn’t see any mention on the ethics of data privacy risk from these new technologies. You have done a really good job at reviewing so many articles, but not a single article on the ethics and risk. There are recent articles on this topic that reviews recent and relevant literature, for example, on the related topic of ‘Ethics and Shared Responsibility in Health Policy’ - see: https://doi.org/10.3390/su13158355 It would be interesting to see a few sentences review and comparison of your work in relations to these recent studies in related topics.

- One final comment, you should check if all the things discussed in the article, are also discussed in the conclusion. because the conclusion seems a bit short on key findings and conclusions (probably the table/figure mentioned in the previous comment will also help you clarify the key findings). If you think you have covered everything, that’s OK, but just to mention that conclusion is the best chapter to outline your key findings and key conclusions.

Otherwise, well done and I am looking forward to reading the revised version. 

seems OK, but I am not a native English speaker. 

Author Response

- I have finished reading the article and I didn’t see any mention on the ethics of data privacy risk from these new technologies. You have done a really good job at reviewing so many articles, but not a single article on the ethics and risk. There are recent articles on this topic that reviews recent and relevant literature, for example, on the related topic of ‘Ethics and Shared Responsibility in Health Policy’ - see: https://doi.org/10.3390/su13158355 It would be interesting to see a few sentences review and comparison of your work in relations to these recent studies in related topics.

Dear reviewer, thank you for your comment. We discussed ethical concerns in the “current limitation and future development” in the revised version of the manuscript (page 14, lines 604-618).

- One final comment, you should check if all the things discussed in the article, are also discussed in the conclusion. because the conclusion seems a bit short on key findings and conclusions (probably the table/figure mentioned in the previous comment will also help you clarify the key findings). If you think you have covered everything, that’s OK, but just to mention that conclusion is the best chapter to outline your key findings and key conclusions.

 Thank you for pointing this out. We expanded and improved the “discussion” section highlighting the key point discussed in the review.

Round 2

Reviewer 2 Report

The authors have addressed all our concerns

Thanks